# Enhanced Ferroelectric, Dielectric Properties of Fe-Doped PMN-PT Thin Films

**DOI:** 10.3390/nano11113043

**Published:** 2021-11-12

**Authors:** Chao Feng, Tong Liu, Xinyu Bu, Shifeng Huang

**Affiliations:** Shandong Provincial Key Laboratory of Preparation and Measurement of Building Materials, University of Jinan, Jinan 250022, China; mse_fengc@ujn.edu.cn (C.F.); shandongmems@163.com (T.L.); buxinyu98@163.com (X.B.)

**Keywords:** PMN-PT thin films, preferred orientation, ferroelectric property, dielectric property

## Abstract

Fe-doped 0.71Pb(Mg_1/3_Nb_2/3_)O_3_-0.29PbTiO_3_ (PMN-PT) thin films were grown in Pt/Ti/SiO_2_/Si substrate by a chemical solution deposition method. Effects of the annealing temperature and doping concentration on the crystallinity, microstructure, ferroelectric and dielectric properties of thin film were investigated. High (111) preferred orientation and density columnar structure were achieved in the 2% Fe-doped PMN-PT thin film annealed at 650 °C. The preferred orientation was transferred to a random orientation as the doping concentration increased. A 2% Fe-doped PMN-PT thin film showed the effectively reduced leakage current density, which was due to the fact that the oxygen vacancies were effectively restricted and a transition of Ti^4+^ to Ti^3+^ was prevented. The optimal ferroelectric properties of 2% Fe-doped PMN-PT thin film annealed at 650 °C were identified with slim polarization-applied field loops, high saturation polarization (*P*_s_ = 78.8 µC/cm^2^), remanent polarization (*P*_r_ = 23.1 µC/cm^2^) and low coercive voltage (*E*_c_ = 100 kV/cm). Moreover, the 2% Fe-doped PMN-PT thin film annealed at 650 °C showed an excellent dielectric performance with a high dielectric constant (*ε*_r_ ~1300 at 1 kHz).

## 1. Introduction

Lead-based materials have been researched for a series of device applications, such as actuators, non-volatile memories, transducers and sensors [1,2,3,4]. The Pb(Mg_1/3_Nb_2/3_)O_3_-PbTiO_3_ (PMN-PT), as relaxor ferroelectrics, with outstanding dielectric, ferroelectric and piezoelectric properties, has been widely investigated in terms of bulk ceramics and single crystals [5,6,7,8]. However, the single crystals and ceramics have been unable to meet the requirements of integrated and miniaturized devices with the development of micro-electromechanical systems (MEMS) in recent years. Advances have been made in synthesis and modification of the PMN-PT thin films because of unique advantages, such as low synthesis temperature, small size, easy integration [9,10]. However, it is still a big challenging to prepare PMN-PT thin films with single phase, crack-free and dense microstructure. In addition, the performances of thin films are far weaker than the bulk materials because of the thickness effect of film and the clamping effect of the substrates.

In order to improve the quality of crystallization and enhance electrical properties of PMN-PT thin film, researchers have done a great deal of works on topics including site engineering, regulating of annealing process and so on. For example, rare earth element doping can effectively enhance electrical properties of PMN-PT thin films [11]; Gabor et al. have reported the process window can be expanded in order to obtain pure phase PMN-PT thin film with the help of LaNiO_3_ layer [12]; Keech et al. prepared (001)-orientation PMN-PT films with a PbO buffer to account for Pb loss [13]; Shen et al. found the additive methanamide can enhance electrical properties of PMN-PT thin films with reduced residual stress of the film and dense microstructure [14]. Deposition methods of various kinds have been reported to synthesize PMN-PT thin films, including pulsed laser deposition, magnetron sputtering, metalorganic chemical vapor deposition and chemical solution deposition (CSD) [15,16,17,18]. CSD is believed to be a further industrial method with easy composition control of precursor solutions, excellent reproducible, low synthesis temperature and low cost. For the CSD route, it is critical to control crystallization through regulating annealing process parameters because annealing treatment is necessary to crystallize the amorphous films. Due to the electrical properties of PMN-PT thin films being strongly dependent on their grain orientation, the realization of preferential orientation is also expected to improve the electrical properties of ferroelectric thin films. Therefore, the synthesis of an ideal PMN-PT thin film should include two aspects: appropriate annealing parameters and the preferential orientation resultant thin film.

In this research, 0.71Pb(Mg_1/3_Nb_2/3_)O_3_-0.29PbTiO_3_ thin films, with different doping concentrations of the Fe element used as acceptor doping, were prepared on Pt(111)/Ti/SiO_2_/Si substrate by the CSD method. The effects of annealing temperature and doping concentration on the phase, microstructure, dielectric and ferroelectric properties of PMN-PT thin films were systematically researched. The highly (111) preferred orientation and dense microstructure were achieved in the 2% Fe-doped PMN-PT thin film annealed at 650 °C, which lead to the enhanced ferroelectric with high saturation polarization (*P*_s_ = 78.8 µC/cm^2^) and remanent polarization (*P*_r_ = 23.1 µC/cm^2^) as well as dielectric with a high dielectric constant (*ε*_r_~1300 at 1 kHz).

## 2. Materials and Methods

The precursor solutions of 0.71Pb(Mg_1/3_Nb_2/3_)O_3_-(0.29-x%)PbTiO_3_-x%PbFeO_3_ (PMN-PT-xFe, x = 0, 2, 4, 8) thin films were prepared by the CSD method. Trihydrated lead acetate, magnesium ethoxide, niobium ethoxide, titanium iso-propoxide and iron nitrate were used as the raw materials. The solvent was 2-methoxyethanol and acetic acid; about 5 mol% excess of lead was added to compensate the volatilization during heat treatment. The solution was stirred using a magnetic stirrer for 12 h, and further aged for 24 h to promote the uniformity of solution. The concentration of precursor solutions was adjusted to be 0.2 M; the precursor was spin-coated on Pt(111)/Ti/SiO_2_/Si substrate at 3000 rpm for 30 s. The obtained film was heated at 350 °C for 2 min for solvents volatilizing and amorphous films formation, and subsequently annealed at the specified annealing temperature (600~700 °C) for 5 min in air atmosphere for film densification and crystallization in a programmed rapid thermal annealing furnace. The desirable thickness was achieved by a 12 time layer-by-layer spin-coating and annealing processes with one layer thickness of about 25 nm.

Crystalline phases were identified by X-ray diffraction (XRD) measurement using Bruker D8 Advanced XRD (Berlin, Germany). The microstructures of thin films were recorded using a field emission scanning electron microscopy (SEM) and atomic force microscope (AFM, MFP-3D Origin+, Concord. MA, USA). The ferroelectric tester (TF Analyzer 3000, Aachen, Germany) was used for polarization-electric field (*P*–*E*) hysteresis loops and leakage current density of thin films. The relative permittivity (*ε*_r_) and dielectric loss (tanδ) were measured using an impedance analyzer (Agilent 4294A, Santa Clara, CA, USA).

## 3. Results and Discussion

Figure 1a shows the XRD patterns of PMN-PT-2Fe thin films annealed at different temperatures. It can be seen that the PMN-PT perovskite phase coexists with a pyrochlore phase (peak at 2θ = 29.3°) in thin films treated over the entire temperature range. The larger fraction of pyrochlore phase is attributed to the lead remaining due to the inadequate annealing. The decrease of concentration for pyrochlore phase from 4.2% for 600 °C to 1.4% for 700 °C indicates the phase transition from pyrochlore to perovskite phase. Moreover, the pyrochlore phase almost disappeared which is due to fuller annealing when the annealing temperature reached 700 °C. In addition, it is worth noting that the preferred orientation of thin films also depends on the annealing temperature; the degree of preferred orientation is defined by the preferential orientation parameter *α*_hkl_ calculated with the following equation:(1)$$\alpha_{\mathrm{hkl}}=\frac{I_{\mathrm{hkl}}}{\sum I_{\mathrm{hkl}}}$$
where, *I*_hkl_ is the peak intensity of (hkl) orientation peak of thin film. The *α*_111_ of PMN-PT-2Fe thin films with the annealing temperature of 600, 650 and 700 °C were 76%, 90% and 74%, respectively. The highly (111) preferred orientation will lead to the excellent electrical performances of PMN-PT thin films [19]. The crystallization of PMN-PT-xFe thin films as a function of doping concentration of Fe element is shown in Figure 1b. The obviously enhanced intensity of peak indicates that the introduction of Fe ion promotes the crystallization of thin film. The preferred orientation transition from (111) to (110) was observed when the doping concentration of Fe ion increased from 2% to 8%. The degree of (110) preferential orientation of PMN-PT-8Fe was up to 49%. The preferred orientation of ferroelectric thin films is subject to three factors: (i) template layer [20], (ii) formation of intermetallic phase during film crystallization [21], (iii) crystallization kinetic [22]. The (111) preferred orientation of the PMN-PT and PMN-PT-2Fe thin films is attributed to the template effect of the Pt(111) substrate [23]. However, continuing to increase the doping concentration will impact the preheat decomposition of the sol, which will change the crystal orientation [24]. When the doping concentration is up to 4%, the preferred orientation transfer from (111) to (110), indicating that effect on the preheat decomposition surpasses the template effect.

Figure 2 shows the microstructures of PMN-PT-2Fe thin films annealed at different temperatures; the surface of the film, annealed at 600 °C, showed an abundance of very small features, which is due to the incomplete growth of the grain. It is clear to observe that the grains grew as the annealing temperature increased, and that the film annealed at 700 °C had the largest grains, at 100–250 nm. The uniform, dense grains, as well as the crack-free thin film were achieved at an annealing temperature of 650 °C, which is beneficial for improving electrical properties. However, the obvious cracks appeared in the film annealed at 600 and 700 °C, which leads to a deterioration in performance. As shown in cross-sectional SEM images, all films are composed of relatively dense microstructures. The PMN-PT-2Fe thin film annealed at 600 °C displays a layered structure. The typical columnar structure can be observed in the film annealed at 650 °C, which corresponds to the preferred orientation growth, as shown in XRD patterns in Figure 1a. However, the continuous increase of annealing temperature does not lead to further optimization of the column structure, accompanied by the appearance of voids. The thickness of all thin films is about 290 nm. The AFM images of PMN-PT-2Fe thin films annealed at different temperatures in the scanning area of 10 μm × 10 μm were shown in Figure 2g–i, representing the root square roughness is 4.914 nm, 3.061 nm, 8.089 nm for 600, 650 and 700 °C, respectively.

Figure 3 shows the microstructures of PMN-PT-xFe thin films with different Fe doping concentrations. It can be seen that the fine grain was achieved in the pure PMN-PT thin film. The larger grains developed as the doping concentration of Fe ion increased. However, obvious voids can be observed between layers in thin films with 4% and 8% Fe doping, although the columnar structure still exists, which is consistent with the weakening of preferred orientation, as shown in Figure 1b. Figure 3i–l shows the AFM images of PMN-PT-xFe thin films with different Fe doping concentrations, representing the root square roughness as 3.776 nm, 3.061 nm, 3.999 nm, 4.077 nm for 0, 2, 4 and 8% doping concentrations, respectively.

Figure 4a shows the leakage current density–voltage (*J–V*) curves of PMN-PT-2Fe thin films as a function of annealing temperature from 600 to 700 °C. It can be seen that the curves of all thin films exhibit asymmetry between positive and negative applied voltage, which might be attributed to the different interface states between top Au/PMN-PT-2Fe thin film and bottom Pt/PMN-PT-2Fe thin film. The leakage current density exhibits steep increasing tendency in the low voltage, which suggests that the dominant conduction mechanism is Ohmic. In the high voltage scale, the leakage current densities increase slightly, which is attributed to the contribution of space charge limited current conduction mechanism. A sudden increase in leakage current for PMN-PT-2Fe thin film annealed at 700 °C was attributed to the abnormal big grain. The shortened grain boundaries accompanied by the big grain provided shorter leakage current path [25].

The effect of doping concentration on the leakage current density of PMN-PT-xFe thin films was measured, as shown in Figure 4b. It can be seen that the pure PMN-PT thin film exhibits the highest leakage current density. The effective decrease leakage current can be obtained in 2% Fe-doped thin film. The leakage current of PMN-PT-based thin films is mainly originated from the following two aspects: (1) oxygen vacancies ($V_{O^{2-}}^{••}$) due to the volatilization of PbO; (2) the transition of Ti^4+^ to Ti^3+^. The mobile $V_{O^{2-}}^{••}$ can be effectively restricted through the formation of $2{\left(Fe_{Ti^{4+}}^{3+}\right)}^{\prime }$-$V_{O^{2-}}^{••}$ [26]. In addition, the introduction of low-valance Fe, as acceptor dopants, will prevent the transition of Ti^4+^ to Ti^3+^ [27]. Among all the films, the PMN-PT-2Fe thin film exhibits the lowest leakage current density of ~10^−3^A cm^−2^ in high voltage, which is more than three orders of magnitude lower than that of the pure PMN-PT thin film. However, continuing to raise the Fe element does not lead to a further decrease of the leakage current. This could be due to the fact that ${\left(Fe_{Ti^{4+}}^{3+}\right)}^{\prime }$ has exceeded the required amount to restrict $V_{O^{2-}}^{••}$. Excess ${\left(Fe_{Ti^{4+}}^{3+}\right)}^{\prime }$ gathered at the grain boundary will act as leakage for the current path, which then deteriorates the leakage characteristic.

Figure 5 displays the *P–E* hysteresis loops of PMN-PT-2Fe thin films annealed at various temperatures, from 600 to 700 °C. The *P–E* hysteresis loops of PMN-PT-2Fe thin films were not well developed at a relatively low temperature (600 °C) because a large amount of non-ferroelectric pyrochlore phase existed in the thin film. With the increasing annealing temperature, the ferroelectric hysteresis loops become slim without leakage characteristics; indeed, the best ferroelectric properties were observed in the PMN-PT-2Fe thin film annealed at 650 °C. High annealing temperature improves the crystallization and reduces pyrochlore phase content. However, the hysteresis loops turn bad again at an annealing temperature of 700 °C, which is due to the more defects induced by the over volatilization with high annealing temperature. Figure 5d shows the *P–E* hysteresis loops of PMN-PT-2Fe thin films at those breakdown strengths. The sharp drop in breakdown strength for PMN-PT-2Fe thin film annealed at 700 °C is due to the significant increase of leakage current. In addition, compared with one annealed at 650 °C, the PMN-PT-2Fe thin film annealed at 700 °C displays a larger coercive electrical field, which is due to the amount of $2{\left(Fe_{Ti^{4+}}^{3+}\right)}^{\prime }$-$V_{O^{2-}}^{••}$. The internal fields were created with the dipoles of $2{\left(Fe_{Ti^{4+}}^{3+}\right)}^{\prime }$-$V_{O^{2-}}^{••}$, which stabilizes the domain configuration.

Figure 6 shows the *P–E* hysteresis loops of PMN-PT-xFe thin films with different Fe doping concentrations. It can be seen that all the doped thin films exhibit improved *P–E* loops compared with pure PMN-PT thin film with a round shaped *P–E* loop [28]. The PMN-PT-2Fe thin film exhibited the so-called slim loops as a feature typical for the relaxor ferroelectrics, which is due to obviously reduced leakage current density. However, this feature disappeared as the doping concentration continuously increased, owing to the excess ${\left(Fe_{Ti^{4+}}^{3+}\right)}^{\prime }$ gathered at the grain boundary, which lead to the deterioration of leakage characteristics. The optimal ferroelectric properties were exhibited in the PMN-PT-2Fe thin film with high saturation polarization (*P*_s_ = 78.8 µC/cm^2^), remanent polarization (*P*_r_ = 23.1 µC/cm^2^) and low coercive voltage (*E*_c_ = 100 kV/cm). At the same time, the asymmetry of hysteresis loops of the sample increases as the doping concentration increases. The increasing asymmetry is attributed to two factors: (i) internal fields created by the dipoles of $2{\left(Fe_{Ti^{4+}}^{3+}\right)}^{\prime }$-$V_{O^{2-}}^{••}$, (ii) accumulation effect of excess Fe ion.

The ferroelectric nature of PMN-PT thin films can be identified from irreversible nature of dielectric performance versus electric field behavior. Voltage dependence of dielectric constant (*ε*_r_-*V*) and loss tangent (tanδ-*V*) of PMN-PT thin films was measured at 100 Hz, as shown in Figure 7. The ferroelectric properties of PMN-PT thin films were confirmed by the butterfly-shaped curve. In order to study the nonlinear dielectric property, the corresponding dielectric tunable performance was calculated. The dielectric tunability (T) is defined as: T = [(*ε*_0_ − *ε*_r_)/*ε*_0_] × 100%, where *ε*_0_ and *ε*_r_ represent the dielectric constant values at zero and the maximum applied field, respectively. According to the *ε*_r_-*V* curves of the PMN-PT-2Fe thin films annealed at different temperatures as shown in Figure 7a, the calculated T values at 10 V are 52%, 73% and 70% for annealing temperature of 600, 650 and 700 °C, respectively. The increase of T with the increase of annealing temperature is attributed to the improved crystallization and grain size of the films, which is corresponding to the XRD and SEM images as shown in Figure 1 and Figure 2 [29,30]. The dielectric loss (tanδ) shows the same tendency as the annealing temperature increases. Figure 7b shows the *ε*_r_-*V* curves of PMN-PT-xFe thin films with different Fe doping concentrations. The calculated T values at 10 V are 34%, 73%, 64% and 46% for PMN-PT-xFe thin films with Fe doping of 0, 2, 4 and 8%, respectively. It can be seen that the T first increases and then decreases with the increase in doping concentration. The decrease in T is attributed to the deterioration of leakage characteristics. In addition, the transition of preferred orientation of PMN-PT thin film from (111) to (110) may be another factor that causes T to change, which corresponds to the previous literatures [31,32]. Moreover, tanδ exhibits a tendency to first decrease before increasing with the increase in doping concentration.

Frequency-dependent *ε*_r_ and tanδ of PMN-PT-2Fe thin films annealed at different temperatures and PMN-PT-xFe thin films with different doping concentrations of Fe element were measured from 1 kHz to 1 MHz at room temperature, as shown in Figure 8. It can be observed that the annealing temperature and doping concentration have a strong effect on its dielectric property. For all samples, *ε*_r_ decreased gradually with the increase in frequency, which is due to the fact that the polarization process of some frameworks (such as space charges) cannot be achieved [33,34]. The significantly low *ε*_r_ of PMN-PT-2Fe thin film annealed at 600 °C is attributed to presence of pyrochlore phase with low *ε*_r_ and smaller grain size of thin film, which are also evident form XRD patterns and surface SEM images [35]. The *ε*_r_ of PMN-PT-2Fe thin film increased with the increase of annealing temperature, which is due to the decrease of pyrochlore phase and the growth of crystal grain. However, continuing to raise the Fe element does not lead to a further increase of *ε*_r_. This could be due to the fact that ${\left(Fe_{Ti^{4+}}^{3+}\right)}^{\prime }$ has exceeded the required amount to restrict $V_{O^{2-}}^{••}$. Excess ${\left(Fe_{Ti^{4+}}^{3+}\right)}^{\prime }$ gathered at the grain boundary will act as a leakage current path, then waken *ε*_r_. The PMN-PT-2Fe thin film annealed at 650 °C shown the largest *ε*_r_ of ~1300 which is comparable to those reported in the literature where PMN-PT thin films were grown on different substrates [36,37,38,39]. The tanδ for all samples increased with increasing frequency, which is due to the extrinsic loss and dipolar lagging phenomena [40].

## 4. Conclusions

PMN-PT thin films with different Fe doping concentrations have been synthesized on Pt/Ti/SiO_2_/Si substrate by the CSD technique. The crystallinity, orientation, microstructure and defect dipoles induced by ion substitution were attributed to the origin of the enhanced electrical performances; 2% Fe-doped PMN-PT thin film annealed at 650 °C showed the high (111) preferred orientation, and the preferred orientation transferred to random orientation as the doping concentration increased. The dense columnar structure was obtained in 2% Fe-doped PMN-PT thin film annealed at 650 °C. The excessive annealing temperature and excessive doping concentration will lead to appearance of cracks. In addition, compared with the pure PMN-PT thin film, the effectively enhanced leakage characteristic was obtained in 2% Fe-doped PMN-PT thin film, which is because of the reduction of movable $V_{O^{2-}}^{••}$ concentration and restraint from Ti^4+^ to Ti^3+^. The enhanced ferroelectric (*P*_s_ = 78.8 µC/cm^2^, *P*_r_ = 23.1 µC/cm^2^, *E*_c_ = 100 kV/cm) and dielectric properties (*ε*_r_ ~1300 at 1kHz) have been obtained in 2% Fe-doped PMN-PT thin film annealed at 650 °C. These results provide the important guiding significance for controlling the grain orientation in preparation of ferroelectric thin films and enhancing the electrical performances of ferroelectric thin films.

## Figures and Tables

**Figure 1 nanomaterials-11-03043-f001:**
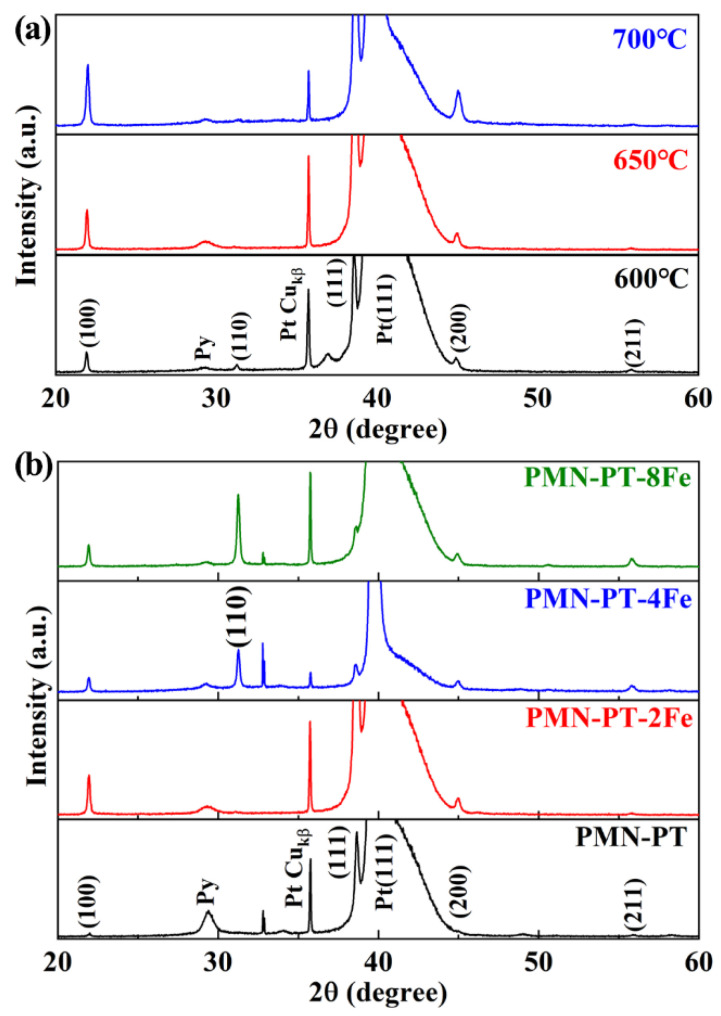
(**a**) XRD patterns of PMN-PT-2Fe thin films annealed at different temperatures, (**b**) XRD patterns of PMN-PT-xFe thin films annealed at 650 °C with different doping concentrations of Fe element. Note that the peaks at 21.95°, 44.98°, 38.65°, 55.81° correspond the (100), (200), (111) and (211) diffraction peaks of PMN-PT thin films, respectively. The 29.39° peak corresponds to the diffraction peak of pyrochlore structure.

**Figure 2 nanomaterials-11-03043-f002:**
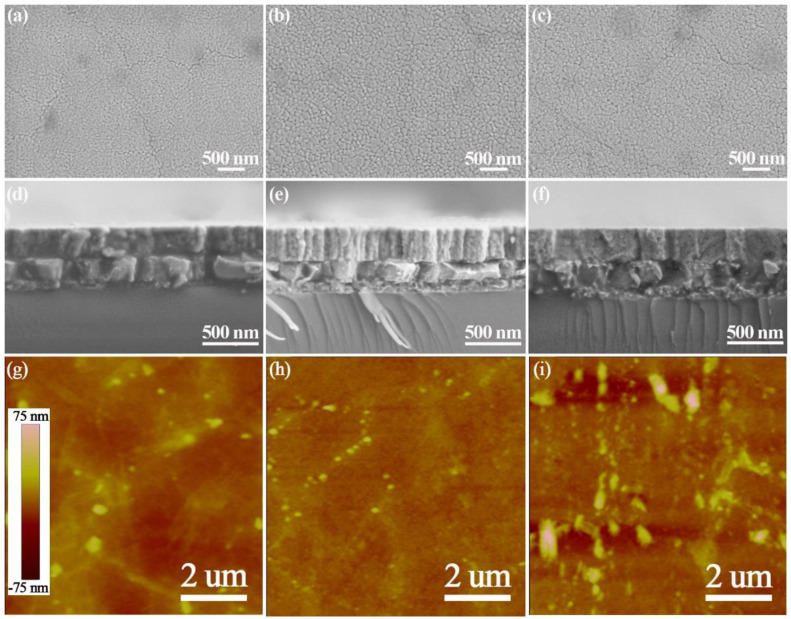
Microstructures of PMN-PT-2Fe thin films annealed at different temperatures. (**a**,**d**,**g**) 600 °C; (**b**,**e**,**h**) 650 °C; (**c**,**f**,**i**) 700 °C. Note the cracks are observed in (**a**,**c**) and the uniform, dense and crack-free structure is obtained in (**b**), the typical columnar structure is shown in (**e**).

**Figure 3 nanomaterials-11-03043-f003:**
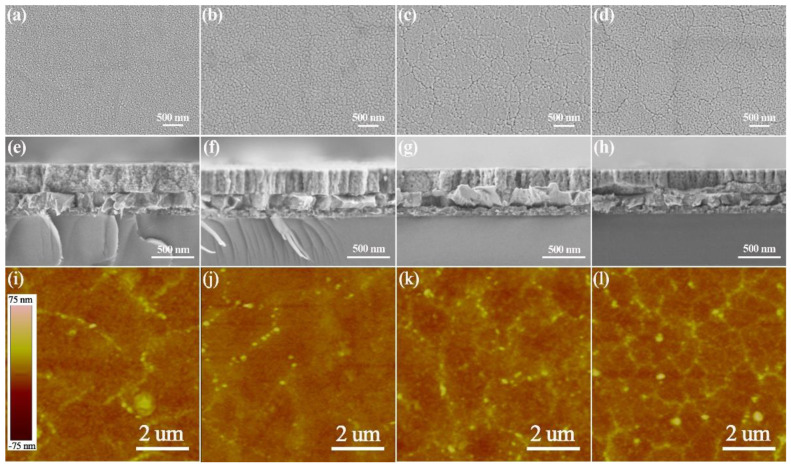
Microstructures of PMN-PT-xFe thin films with different doping concentration of Fe element. (**a**,**e**,**i**) x = 0; (**b**,**f**,**j**) x = 2; (**c**,**g**,**k**) x = 4; (**d**,**h**,**l**) x = 8. Note that the fine grains are obtained in (**a**), the cracks are observed in (**c**,**d**) and the uniform, dense and crack-free structure is obtained in (**b**), the layered structure is shown in (**e**), the typical columnar structure is shown in (**f**,**g**,**h**).

**Figure 4 nanomaterials-11-03043-f004:**
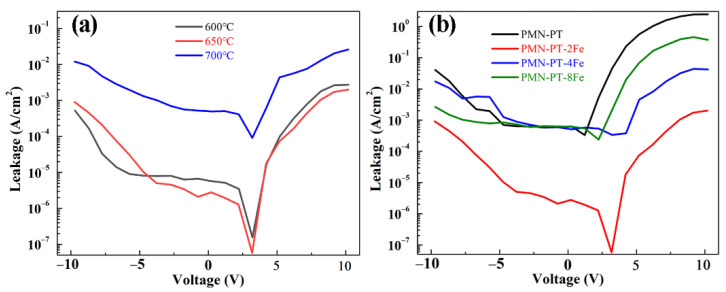
(**a**) Leakage current density of PMN-PT-2Fe thin films annealed at varied temperatures, (**b**) Leakage current density of PMN-PT-xFe thin films with different doping concentrations of Fe element.

**Figure 5 nanomaterials-11-03043-f005:**
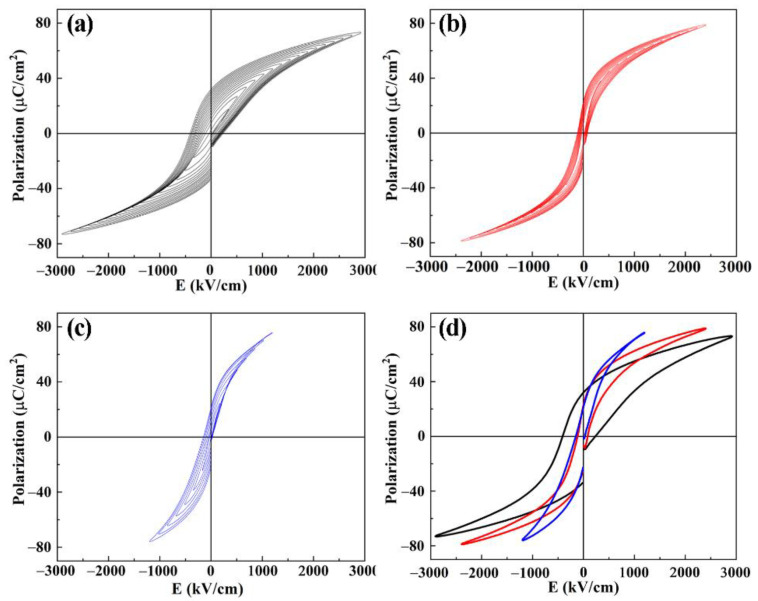
*P–E* hysteresis loops of PMN-PT-2Fe thin films annealed at various temperatures (**a**) 600 °C, (**b**) 650 °C, (**c**) 700 °C, (**d**) *P–E* hysteresis loops at breakdown strength.

**Figure 6 nanomaterials-11-03043-f006:**
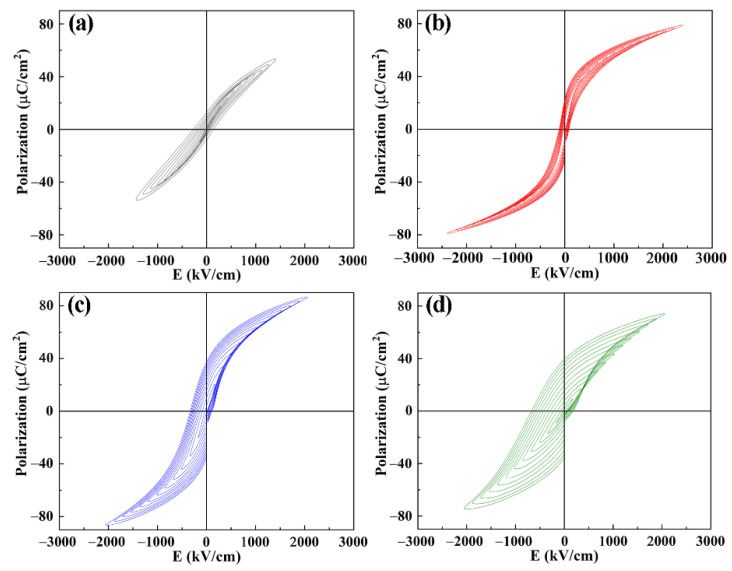
*PE* hysteresis loops of PMN-PT-xFe thin films with different doping concentration of Fe element (**a**) x = 0, (**b**) x = 2, (**c**) x = 4, (**d**) x = 8.

**Figure 7 nanomaterials-11-03043-f007:**
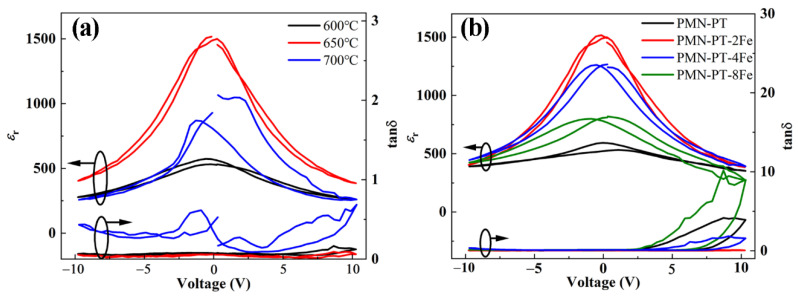
(**a**) Voltage dependence of *ε*_r_ and tanδ for PMN-PT-2Fe thin films annealed at different temperatures, (**b**) Voltage dependence of *ε*_r_ and tanδ for PMN-PT-xFe thin films with different doping concentrations of Fe element.

**Figure 8 nanomaterials-11-03043-f008:**
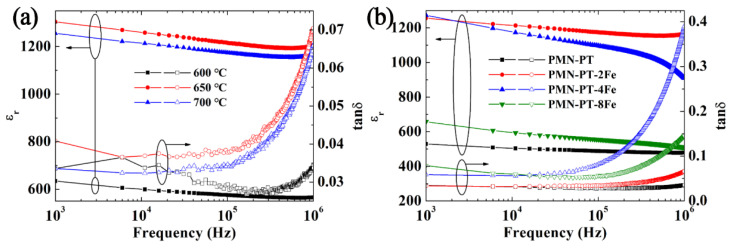
(**a**) The frequency dependence of *ε*_r_ and tanδ of PMN-PT-2Fe thin films annealed at different temperatures, (**b**) The frequency dependence of the *ε*_r_ and tanδ of PMN-PT-xFe thin films with different doping concentrations of Fe element.

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
