# Peer review of "Enhanced Ferroelectric, Dielectric Properties of Fe-Doped PMN-PT Thin Films"

_nanomaterials, 2021, doi:10.3390/nano11113043_

Round 1
Reviewer 1 Report
In general, this paper does not seem to have evident originality compared to previous studies.
Introduction
(1) This section is not sufficiently developed. In contrast to the existing CSD-based PMN-PT thin film research, additional explanations are needed for the originality, the difference, the advanced methods of this research.
Materials and Methods
(1) It is necessary to specify the process sequence and conditions in detail. It may also be a good method to insert additional references to well-known processes or previous studies.
(2) line 66 - What is a desirable thickness? How thick does the spin coating deposit one layer?
Results and Discussion
(1) line 84 - The authors commented that the pyrochlore phase decreased with the annealing temperature. However, it seems to have increased in the range of 600 to 650°C.
(2) Please indicate the annealing temperature for the Fig. 1(b) samples in the text or figure caption.
(3) lines 95~98 - The authors commented that enhanced peak intensity and orientation transition from (111) to (110) were observed with the increase of Fe doping concentration. What is the mechanism? Would you please add detailed descriptions or references for the rise of crystallinity and orientation changes with Fe doping?
(4) In Figs. (5) and (b), the authors are needed to use a similar range of the x- and y-axes. That makes it easier for readers to compare the results.
(5) line 195 - Fig. 7 has no loss tangent data. Please add data or edit the text?
(6) Would you please further present and discuss all figures in the text and provide references where available? The authors need further to discuss their results in a more correlated manner.
Conclusions
(1) It is necessary to emphasize the significance and differentiation of the results obtained through this study in the related fields.
Reviewer 2 Report
The authors tackle a hot topic related to relaxor ferroelectrics of PMN-PT of high interest for energy storage, in particularvthe developing of PMN-PT thin films of high quality. They propose PMN-PT thin films doped with Fe that are obtained by chemical solution deposition and annealed By RTA. They study the effects of RTA temperature and dopant concentration on microstructure, crystalline structure, ferroelectric and dielectric properties, and also on leakage current. The results are very good, comparable with literature.
In my opinion, the manuscript deserves to be published in Nanomaterials.
I have minor comments:
-how much reproducible is CSD method?
-CSD is industrial?
-why doping with Fe?
-the rapid thermal processor characteristics? Lamps distribution, size of susceptor? What RTP model? What about the ramping up and down?
-in Fig. 1 I think that the caption for b should be detailed: peaks position with crystalline structure and association, also specify RTA temperature
-please detail caption of Fig2... SEM images,... Cross-section SEM images,...
-can you detail the description in the text of cross-section SEMs, what is the difference between them?
- please deatail caption of Fig. 3
-can you explain the difference between Fig 3 f and g?
Round 2
Reviewer 1 Report
The authors responded appropriately to all matters.
Therefore, I would like to evaluate this manuscript as accepted for publication.